# Metabolism-Related Gene Expression in Circulating Tumor Cells from Patients with Early Stage Non-Small Cell Lung Cancer

**DOI:** 10.3390/cancers14133237

**Published:** 2022-06-30

**Authors:** A. Zafeiriadou, I. Kollias, T. Londra, E. Tsaroucha, V. Georgoulias, A. Kotsakis, E. Lianidou, A. Markou

**Affiliations:** 1Analysis of Circulating Tumor Cells Lab, Lab of Analytical Chemistry, Department of Chemistry, National and Kapodistrian University of Athens, 15771 Athens, Greece; anza.chem@gmail.com (A.Z.); gkollias286@yahoo.gr (I.K.); doralontra@hotmail.gr (T.L.); lianidou@chem.uoa.gr (E.L.); 2‘Sotiria’ General Hospital for Chest Diseases, 11527 Athens, Greece; emilygeola@yahoo.gr; 3First Department of Medical Oncology, IASO General Hospital of Athens, 15123 Athens, Greece; georgulv@otenet.gr; 4Department of Medical Oncology, University General Hospital of Larissa, 41334 Larissa, Greece; thankotsakis@hotmail.com

**Keywords:** circulating tumor cells, glycolysis, Warburg effect, molecular characterization, epithelial markers, EMT

## Abstract

**Simple Summary:**

In the present study, the expression of three Metabolism-Related Enzymes (MRGs) that are related to glucose and pyruvate metabolism, in parallel with glucose and monocarboxylate transporter expression (*HK2*, *MCT1*, *PHGDH*), was studied in CTCs isolated from the peripheral blood of early stage NSCLC patients at different timepoints. The expression levels of all tested MRGs decreased in CTCs one month after surgery, but a significant increase was noticed at the time of relapse for *PHGDH* and *MCT1* only. An overexpression of MRGs was observed at a high frequency in the CTCs isolated from early NSCLC patients, thereby supporting the role of MRGs in metastatic processes. The glycolytic and mesenchymal subpopulation of CTCs was significantly predominant compared to CTCs that wereglycolytic but not mesenchymal-like. Our data indicate that MRGs merit further evaluation through large and well-defined cohort studies.

**Abstract:**

Purpose: Metabolic reprogramming is now characterized as one of the core hallmarks of cancer, and it has already been shown that the altered genomic profile of metabolically rewired cancer cells can give valuable information. In this study, we quantified three Metabolism-Related Gene (MRG) transcripts in the circulating tumor cells (CTCs) of early stage NSCLC patients and evaluated their associations with epithelial and EMT markers. Experimental Design: We first developed and analytically validated highly sensitive RT-qPCR assays for the quantification of *HK2*, *MCT1* and *PHGDH* transcripts, and further studied the expression of MRGs in CTCs that were isolated using a size-dependent microfluidic device (Parsortix, Angle) from the peripheral blood of: (a) 46 NSCLC patients at baseline, (b) 39/46 of these patients one month after surgery, (c) 10/46 patients at relapse and (d) 10 pairs of cancerous and adjacent non-cancerous FFPE tissues from the same NSCLC patients. Epithelial and EMT markers were also evaluated. Results: *MCT1* and *HK2* were differentially expressed between HD and NSCLC patients. An overexpression of *MCT1* was detected in 15/46 (32.6%) and 3/10 (30%) patients at baseline and at progression disease (PD), respectively, whereas an overexpression of *HK2* was detected in 30.4% and 0% of CTCs in the same group of samples. The expression levels of all tested MRGs decreased in CTCs one month after surgery, but a significant increase was noticed at the time of relapse for *PHGDH* and *MCT1* only. The expression levels of *HK2* and *MCT1* were associated with the overexpression of mesenchymal markers (*TWIST-1* and *VIM*). Conclusion: An overexpression of MRGs was observed at a high frequency in the CTCs isolated from early NSCLC patients, thereby supporting the role of MRGs in metastatic processes. The glycolytic and mesenchymal subpopulation of CTCs was significantly predominant compared to CTCs that were glycolytic but not mesenchymal-like. Our data indicate that MRGs merit further evaluation through large and well-defined cohort studies.

## 1. Introduction

Lung cancer constitutes the second most frequent type of cancer worldwide and still remains the leading cause of cancer deaths for both men and women [1]. NSCLC accounts for 85% of all lung cancer cases and is often diagnosed at an advanced stage, which explains the high mortality rate of the disease [2]. While lung cancer therapies have shown great progress with the discovery of various targeted therapies [3] and the efficient application of immunotherapy for some groups of patients, the current methods used for disease monitoring and therapeutic regimens lack the ability to detect relapse early [4].

Liquid biopsy, a minimally invasive blood-based approach, provides the potential for monitoring a tumor’s evolution in real time, whereas the analysis of tumor-derived factors in the bloodstream already has several clinical applications in early diagnosis, treatment selection and response, the detection of metastasis [5] and even in the monitoring of minimal residual disease (MRD) [6]. Circulating tumor cells (CTCs) and circulating tumor DNA (ctDNA) are the most well characterized liquid biopsy components and their prognostic significance has been documented in many types of cancer, including breast, prostate, lung and colorectal [7].

CellSearch is the only FDA-cleared assay for the enumeration of CTCs in metastatic breast, prostate and colorectal cancer [8,9]. The identification and the enumeration of CTCs in NSCLC patients is limited [10] due to the EMT process [11].

The molecular characterization of CTCs, both at a bulk and single cell level, gives important information for the evolution of cancer [12,13], metastatic processes, the identification of new treatment predictive markers and finally the stratification of patients into prognostic groups [14,15,16].

Metabolic reprogramming has been presented as an emerging hallmark of cancer ever since 2011 [17] and is now considered one of the core hallmarks of cancer [18]. Enhanced aerobic glycolysis accompanied by the capacity of cancer cells to metabolize glucose at an elevated rate constitute the Warburg effect, which is one of the major metabolic pathways observed in cancer cells [19]. The Warburg effect actually comprises a multifaceted collection of causative changes in gene expression consisting of the elevated expression of: (i) glucose transporters; (ii) hexokinase; (iii) pyruvate kinase muscle (PKM2); (iv) pyruvate dehydrogenase kinase (PDK); and (v) specific transcription factors [20]. A great number of enzymes that participate in cell metabolism have been studied as prognostic markers, such as HK2 [21], MCT1 [22,23,24] and PHGDH [25,26], some of which are attractive new therapeutic targets such as MCT1 [27].

Our lab was the first to demonstrate an overexpression of *MCT4* in the CTCs of early stage NSCLC patients. More specifically, we have shown the prognostic significance of *MCT4*, investigating its role as a potential non-invasive tumor biomarker [28]. As the metabolic phenotyping of CTCs has already been evaluated [29,30,31] and as several groups have used suchmetabolic features for CTC isolation [32,33,34], in the present study, we focused on accessing the gene expression (mRNA) of three crucial genes that are associated with different metabolic pathways in the bulk of the size-based CTC fractionsfor early stage NSCLC patients. For this purpose, three Metabolism-Related genes (MRGs)—*HK2*, *PHGDH* and *MCT1*—were carefully chosen, and highly sensitive RT-qPCR assays were developed and validated for their relative quantification. The expression levels of these MRGs were further investigated in size-based CTC fractions at different timepoints of disease and their associations with epithelial and EMT markers were also evaluated. Our findings indicate that markers associated with metabolic function are overexpressed in CTCs and should be prospectively evaluated as non-invasive circulating tumor biomarkers in a large and well-defined cohort of patients with NSCLC.

## 2. Materials and Methods

### 2.1. Clinical Samples

Forty-six patients with early stage NSCLC were enrolled in the study. From these patients, 95 peripheral blood samples (25 mL in EDTA tubes) were prospectively collected; 46 samples were obtained at baseline (pre-surgery), 39 samples were obtained one month after surgery and 10 samples were obtained at the time of relapse. The peripheral blood samples from 19 healthy donors (HDs) were used as controls. All patients gave written informed consent to participate in the study, which was approved by the Ethics and Scientific Committee of Thoracic Diseases General Hospital Sotiria.

Additionally, 10 fresh frozen tissues and corresponding adjacent non-neoplastic tissues were analyzed. All fresh frozen tissues were diagnosed with operable (stage I–III) NSCLC; the tumor histology included squamous cell carcinoma (SCC; *n* = 3) and adenocarcinoma (ADC; *n* = 7) subtypes. In this group, the majority of patients (60%) were smokers and their median age was 58 years. All patients were treatment-naïve when the samples were collected.

All HDs had no known illness or fever at the time of blood draw, no history of malignant disease, were ≥ 35 years old, 52.6% were females and 47.4% were males. The main patient characteristics and their correlation with the expression of MRGs are summarized in Table 1.

### 2.2. CTC Enrichment Using the Parsortix Size-Based Microfluidic Device

CTCs were enriched from a 25 mL sample of whole blood using the EpCAM-independent microfluidic device Parsortix (ANGLE plc, Surrey, UK). A microscope chip was used for the separation of blood components [35,36]. The isolation of total RNA from the enriched CTCs was carried out as previously described [28]. CTCs isolated using the microfluidic device are not 100% pure. Since the presence of co-isolated PBMCs in CTC fractions could affect the specificity of the assays, we evaluated this ‘background noise’ of leucocytes by analyzing peripheral blood samples from 19 HD, employing exactly the same method as was used for the patients. We estimated the cut-off values based on the normalized expression of each gene with respect to *B2M* expression in this control group. Using this approach, we defined a sample as positive for the overexpression of each gene based on the fold change of gene expression in the CTC fraction with respect to the corresponding fraction in the control group.

### 2.3. RNA Extraction from Fresh Frozen Tissues

Tissue sections with a tumor cell percentage of more than 80% were used for RNA extraction. The Qiagen RNeasy Mini Reagent kit (Qiagen, Hilden, Germany) was used to isolate the total cellular RNA according to the manufacturer’s instructions.

### 2.4. RT-qPCR

An in-silico study was performed in order to design the specific primers and TaqMan probes for*MCT1*, *PHGDH* and *HK2* using Primer Premier 5.0 software and the BLAST analysis. The sequences of primers and probes are available in Appendix A.

The expression levels of *CK8, CK18, CK19, TWIST1* and *B2M* (reference gene) were also evaluated using our previously described assays [28].

### 2.5. Optimization of Experimental Conditions

A LightCycler^®^ 480 instrument (Roche, Munich, Germany) was used in order to perform RT-qPCR. Several optimization experiments were carried out in order to validate each assay (results not shown). One positive (MCF-7 cell line) and one negative control was used in each experiment. Genomic DNA at high concentrations were used as templates to ensure that the amplification of gDNA was completely avoided. Finally, *B2M* was used as a reference gene for RT-qPCR.

### 2.6. Statistical Analysis

The chi-square test of independence and the Mann–Whitney test were used to compare the different groups. The Kruskal–Wallis (non-parametric) test was used to test whether the median expression levels were the same between the two different groups. All statistical tests were two sided, and *p*-values less than 0.05 were considered statistically significant. RT-qPCR expression data for each gene were normalized with respect to the reference gene expression in each sample using the 2^−ΔΔCt^ approach [37]. For the evaluation of background noise (co-isolate PBMCs) peripheral blood samples from 19 HD were evaluated in exactly the same way as the patients’ samples. The definitions of positive and negative gene expression were performed based on our previous study [28]. To further increase the specificity of the assays we developed, for every marker showing a gene expression background in the healthy donor samples, a cut-off threshold value was calculated. This was executed by adding the twofold standard deviation to the mean foldchange value of these “false-positive” control samples. Statistical analyses were performed using SPSS Statistics 26.0 (IBM Corp, Armonk, NY, USA).

## 3. Results

The outline of the study is shown in Figure 1.

### 3.1. TCGA Analysis

A thorough search of the literature was conducted in order to select the metabolism-related genes that seem to contribute importantly to the Warburg Effect, especially in NSCLC cases. The TCGA analysis was conducted through the GEPIA web server (http://gepia.cancer-pku.cn/, accessed on 1 February 2022) in order to verify the mRNA expression of the genes in lung adenocarcinoma (LUAD) and lung squamous cell carcinoma (LUSC) tissue samples compared to normal tissues. A plot of gene expression levels between cancerous and normal tissues was available through the GEPIA web server and can be seen in Appendix A. The analysis showed that *PHGDH, HK2* and *MCT1* exhibit statistically significant higher expression levels in cancerous tissues compared to normal tissues in LUSC, which is another important indicator that these genes should be tested in the liquid biopsy setting.

### 3.2. MRG Expression in NSCLC Paired Tissues

The expression levels of each gene were firstly investigated in 10 pairs of NSCLC tissues and their adjacent non-cancerous tissues. We observed that *HK2* was elevated in 6/10 (60.0%) NSCLC tissues and its relative expression (expressed as ΔCt) was significantly higher (*p* = 0.019) in the tumor tissues (median = 6.57) compared to their corresponding non-cancerous tissues (median = 8.24) (Figure 2). Additionally, *PHGDH* was overexpressed in the vast majority of samples (9/10 (90%)),andthe difference between the tumor tissues (median = 6.23) and the adjacent tissues (median = 8.86) was statistically significant (*p* = 0.009) (Figure 2). Finally, *MCT1* was overexpressed in 6/10 NSCLC samples (60%); however, the difference between the tumor tissues (median = 7.07) and the adjacent tissues (median = 9.51) was marginally statistically significant (*p* = 0.052) (Figure 2).

### 3.3. MRG Expression in the CTC Fraction of HD and NSCLC Patients

Overexpression of at least one of the tested genes was detected in 22/46 (47.8%) patient samples before surgery. More specifically, *MCT1* overexpression was detected in 15/46 (32.6%) patients at baseline, *PHGDH* overexpression was detected in 7/46 (15.2%) patients and *HK2* overexpression was detected in 14/46 (30.4%) samples. Our evaluation of the differences in the expression levels revealed a significantly higher expression of *MCT1* and *HK2* (*p* = 0.005 and *p* = 0.015, respectively) in the CTC fraction between HD and early stage NSCLC patients, whereas no statistically significant difference was observed for *PHGDH* (Figure 3). The Cq values of each gene tested at baseline is presented in Appendix A. Our results are based on bulk CTC analysis, so by using a size-based microfluidic device, we are not only enriching our samples with CTCs but we are also co-isolating a low fraction of non-specific PBMCs. The presence of these non-specific cells was verified by the expression of *B2M* in all of our size-based CTC fractions. *B2M* expression was used as an internal control for sample quality in order to avoid false-negative results. Furthermore, *B2M* was used as a reference gene for relative quantification because it is expressed in all cells (both CTCs and PBMCs). Following this procedure and the identical analysis of HD peripheral blood samples, we defined a sample as CTC-positive based on the expression of the specific genes that differentiate CTCs from PBMCs.

Patients with a positive*MCT1* expression were less frequently found to be non-smokers than smokers (*p* = 0.017). No further significant associations were observed between *MRG* expression and clinicopathological parameters.

Tissue samples were not matched. However, as a general consideration, we found a relative fold change in the expression of MCT1 of 3.18 in tumor vs. adjacent tissue and 2.87 in CTC vs. healthy donor tissue. Furthermore, we found a relative fold change in the expression of HK2 of 6.65 in tumor vs. adjacent tissue and 5.49 in CTC vs. healthy donor tissue. Finally, we found a relative fold change in the expression of PHGDH of 7.40 in tumor vs. adjacent tissue and 1.37 in CTC vs. healthy donor tissue.

### 3.4. MRG Overexpression in CTC Fraction at Different Timepoints

For a subgroup of these patients, peripheral blood samples were available both at baseline and one month later (*n* = 39) and at the time of relapse (*n* = 10). One month after surgery, the expression levels of HK2 and MCT1 genes were reduced compared to baseline expression levels. Overexpression was detected for *HK2* in 1/39 CTC samples (2.6%), for *PHGDH* in 4/39 (10.3%) CTC samples and for *MCT1* in 7/39 (17.9%) CTC samples. It is important to mention that the expression levels between these two timepoints differed significantly for *HK2* and *MCT1* (Figure 3).

Moreover, during the follow-up period, 10 patients (21.7%) developed metastases. In this group, *PHGDH* and *MCT1* overexpression was observed in 3/10 (30%) CTC fraction samples, whereas the overexpression of *HK2* was not observed (Figure 3).

The expression of epithelial and mesenchymal markers was also estimated from the same samples containing cDNA. The following heat map (Figure 4) depicts significant heterogeneity in gene expression in the CTC enriched fraction samples from NSCLC patients. Epithelial markers (at least one; *CK-8*, and/or *CK-18*, and/or *CK-19*) were detected in 22/46 (47.8%) samples, whereas the expression of mesenchymal/EMT markers (at least one; *VIM*, and/or *TWIST-1*) was detected in 28/46 (60.9%).The expression levels of *MCT1* and *HK2* were associated with overexpression of the mesenchymal markers *TWIST-1* and *VIM* (*p* = 0.02 for the *MCT1* positive samples and *p* = 0.015 for the *HK2* positive samples). These findings indicate that the number of CTC samples showing both a glycolytic and mesenchymal-like phenotype simultaneously is significantly higher than that of CTC samples which are glycolytic but not mesenchymal-like.

CTCf ractions were considered glycolytic if they showed overexpression of at least one of the *HK2* or *MCT1* genes. CTC fractions that overexpressed at least one of the *VIM* or *TWIST* genes were considered mesenchymal-like. On the other hand, no association between the expression of either *MCT1* or *HK2* and at least one of the epithelial markers (*CK8, CK18, CK19*) was observed. The expression levels of *PHGDH* were not associated with any of the above-mentioned epithelial or mesenchymal markers. Due to the heterogeneity of gene expression in CTCs, no overexpression of either epithelial or mesenchymal markers in the same sample does not necessarily indicate the absence of CTCs in the size-based isolated CTC fraction.

## 4. Discussion

In the present study, the expression of three enzymes related to glucose and pyruvate metabolism, in parallel with glucose and monocarboxylate transporter expression (*HK2*, *PHGDH* and *MCT1*), was studied in CTCs isolated from the peripheral blood of early stage NSCLC patients at different timepoints. Metabolic reprogramming in cancer is now recognized as one of the core hallmarks of cancer [18]. The first reprogrammed metabolism in cancer was known as the Warburg effect, which focused on the capacity of cancer cells to metabolize glucose at an elevated rate [19]. Glycolysis provides a pool of intermediates available for a plethora of biosynthesis pathways, which are needed for the rapidly proliferating cancer cell [38,39].

Alix-Panabières et al. highlighted the differential expression of genes that regulate energy metabolism in the first cell line (CTC-MCC-41) derived from the metastasis-competent CTCs of a patient with colon cancer [40]. The expression of metabolism-related genes in CTCs from NSCLC patients was first evaluated by our group [28]. In the present study, we evaluated the expression of three metabolism-related genes in CTCs known as: (a) *HK2* which belongs to a family of enzymes that is responsible for the catalysis of the first essential step of glucose metabolism by converting glucose to glucose-6-phosphate (G-6-P), (b) *PHGDH* which catalyzes the oxidation of 3-phosphoglycerate—a glycolysis intermediate product—to 3-phosphohydroxypyruvate, which is equally important for cancer cells as it is known that these cells enhance their anabolic reactions (biosynthesis) and (c) *MCT1* which is a highly important molecule for cellular metabolism and pH regulation as it provides a specific mechanism for the transport of lactate across membranes [41]. The high level of *HK2* expression and activity in glycolytic tumor cells has been revealed in PET imaging [42]. Similarly, *HK2* expression is upregulated in a wide range of cancer types [43,44].

Our group has already shown that molecular assays based on real-time PCR, carried out in nucleic acid material (RNA or genomic DNA) and isolated from size-based CTC fractions [15,28,45] can give valuable information for the molecular characterization of CTCs at the level of gene expression. It is clear that in this approach we are not verifying the presence of CTCs by imaging through immunofluorescence, but through the genetic material isolated from size-based CTC fractions. This approach has also been extensively and successfully used for the molecular characterization of size-based CTC fractions by other research groups [46,47,48]. The microfluidics system that was used in this study assures the depletion of contaminating leucocytes by up to 10^6-^fold, providing a size-based CTC fraction with high purity. Improving the depletion of leukocytes reduces the RT-qPCR background noise and improves both the specificity and sensitivity of the molecular approach [49]. The number of PBMCs co-isolated with CTC from peripheral blood is not exactly the same from patient to patient, but the number of PBMCs trapped non-specifically in the microfluidics chamber is almost stable. By using a considerable number of healthy donor samples, analyzed exactly the same way and using 2SD as a cut-off, we can minimize the effects of the presence of non-specifically co-isolated cells beyond CTCs. Based on this, and on the fact that *B2M* copies are almost the same in CTCs and PBMCs, normalization with *B2M* gives an estimation of target-gene overexpression in CTCs. The microfluidics system PARSORTIX (ANGLE) that was used in our study has recently been given FDA approval for use in CTC isolation(https://finance.yahoo.com/news/angle-receives-fda-clearance-parsortix-120000516.html, accessed on 25 May 2022).

Bioinformatic analyses of the TCGA datasets demonstrated that in LUSC tissues, *PHGDH*, *HK2* and *MCT1* mRNA levels were higher than those in normal lung tissues [26,50,51,52]. Using 10 pairs of NSCLC tissues, we verified that these genes were overexpressed in high levels, ranging from 60% to 90% in cancerous tissues compared to corresponding adjacent non-cancerous tissues. Our findings are consistent with previous studies demonstrating that *HK2* levels were elevated in lung tumors, which has been shown to contribute to the accelerated proliferation of NSCLC cells [53]. Using immunohistochemistry, Tong et al. examined 100 NSCLC tumor tissues and observed high expression of MCT1 in cancer cells [54]. An increased expression of PHGDH at mRNA and protein levels in tumor tissues compared to matched adjacent non-tumor tissues was previously identified [26]. As our results clearly indicated that all tested MRGs are overexpressed in tumor tissue samples compared to adjacent tissues, we proceeded to evaluate their expression levels in the CTCs that shed from the primary tumor. Our analysis showed that both *HK2* and *MCT1* were also found to be overexpressed in the size-based CTC fractions of NSCLC patients. However, according to our findings, there was no concordance between the size-based enriched CTCs and the paired primary tumors with respect to *PHGDH*. Tumor heterogeneity and the limited number of paired tissues are possible explanations for these findings.

By using novel immunofluorescence methods, Kershaw et al. have shown the overexpression of *MCT1* in CTCs enriched in the Veridex™ Cell Search system, demonstrating that the detection of *MCT1* in CTCs could offer the potential to act as a sensitive and selective biomarker assay in the clinical evaluation of drugs targeting this protein [55]. In the current study and for the first time, we studied the expression of *PHGDH*, *MCT1* and *HK2* in CTC fractions isolated from early stage NSCLC patients using size-based EpCAM independent technology (Parsortix). The expression levels of *HK2* and *MCT1* were significantly higher in the CTC fraction from patients than in the corresponding “PBMC” fraction from HD (*p* =0.015 and *p* =0.005, respectively).However, no statistically significant difference was observed for *PHDGH*. These results are in line with other studies that have used immunofluorescence assays [55]. It is important to mention that *HK2* has recently been used as a metabolic-function-associated marker for the identification of CTCs from NSCLC patients and has revealed a HK2high/CKneg CTC population which is not accessible to, and is normally overlooked by, current epithelial-marker-based CTC detection methods [29].

According to our findings, the expression levels of *MCT1* and *HK2* reduced one month after surgery compared to baseline. This is a rational observation resulting from a decrease in CTC counts due to surgical resection of the primary tumor. A possible explanation for the overexpression of these two genes could be that glucose uptake and utilization maybe an early event in NSCLC carcinogenesis. The few CTCs observed after surgery and at the time of relapse may result primarily from appropriate stress management, redox homeostasis and an ability to adapt to the challenging environment instead of utilizing glucose avidly for energy and anabolic processes. This is in accordance with previous studies that have reported increased glucose consumption as an early event in cancer formation for other cancer types [56,57].

The correlation of MRG overexpression in CTCs and patients’ clinicopathological parameters showed an association between the smoking status of patients and *MCT1* overexpression. However, no significant correlation was found between other patients’ clinicopathological parameters and any other MRGs. In a previous study, Domingo-Vidal et al. observed that cigarette smoke induces the metabolic reprogramming of tumor stroma in head and neck squamous cell carcinoma (HNSCC) after culturing cancer-associated fibroblasts in media containing cigarette smoke extracts and co-injecting them with HNSCC cancer cells [58]. Cigarette smoke induced oxidative stress, glycolytic flux and Monocarboxylate Transporter 4 (*MCT4*) upregulation, contributing to tumor aggressiveness. We believe that further investigation of *MCT1* upregulation in NSCLC CTCs and smoking status in a larger cohort of patients is necessary to elucidate possible correlations and to identify any associations with the clinical outcomes of these patients [58].

Interestingly, we observed that the subpopulation of CTCs that exhibited both a glycolytic and mesenchymal phenotype is significantly higher than that of glycolytic but not mesenchymal-like CTCs. This can be explained by the fact that EMT is a trans-differentiation process that requires a high amount of energy and a more rapid biosynthetic rate in order to take place [59]. It has also been shown that lactate, as an end product of aerobic glycolysis, alters the ECM (extracellular matrix) by acidifying it. This is favorable for invasion, which leads to the formation of CTCs [60]. With this assumption in mind, we can also consider that cancer cells expressing mesenchymal surface markers and exhibiting a Warburgian metabolic phenotype are more likely to enter the circulation by intravasating than those with an epithelial phenotype and are therefore more tightly connected to each other. Rivello et al. isolated highly metabolically active cells from the tumor microenvironment of prostate cancer patients based on pH measurements. Interestingly, they observed that the subpopulation of EpCAM^−^ highly metabolic cells was higher than that of EpCAM^+^ highly metabolic cells [61]. This is in line with our results. We isolated and observed a predominant glycolytic and mesenchymal-like subpopulation of CTC fractions. Furthermore, Yang et al. also reported that CTCs with a HK2pos/CKneg phenotype were prevalent in half of the NSCLC patient samples [29], whereas Turetta et al. reported the presence of glucose-avid CTCs in 85% of NSCLC patients [31].

Considering the importance of metabolic processes in cancer, we strongly believe that it is necessary to validate these results in a prospective study, since most studies so far have investigated MRG expression in paired fresh frozen tissues.

## 5. Conclusions

MRG overexpression was observed at a high frequency in the CTCs of early NSCLC patients, supporting its role in metastatic processes. The glycolytic and mesenchymal subpopulation of CTCs was significantly predominant compared to CTCs that were glycolytic and not mesenchymal-like. Our data indicate that MRGs merit further evaluation through large and well-defined cohort studies.

## Figures and Tables

**Figure 1 cancers-14-03237-f001:**
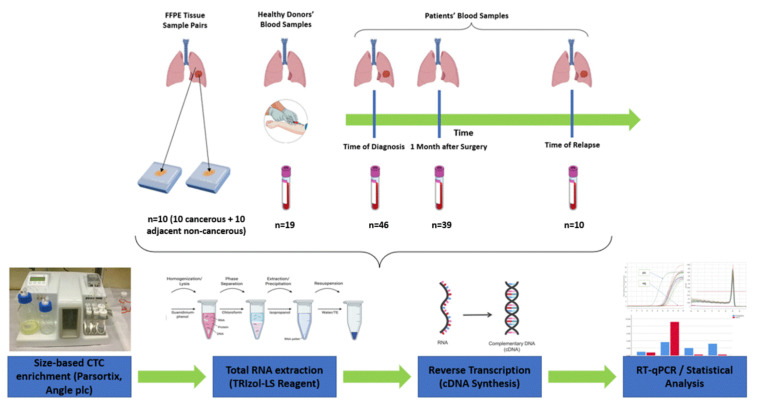
Outline of the experimental procedure.

**Figure 2 cancers-14-03237-f002:**
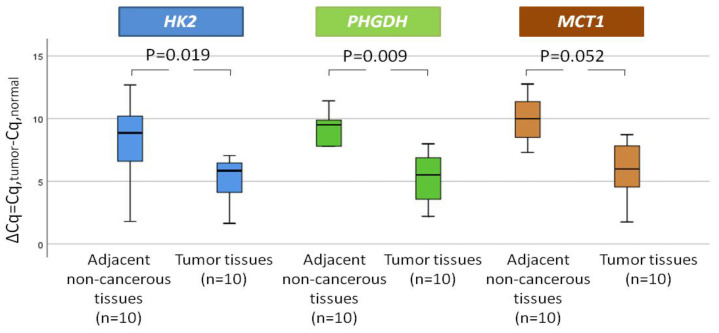
The level of expression of each gene in 10 pairs of NSCLC tissues and their adjacent non-cancerous tissues.

**Figure 3 cancers-14-03237-f003:**
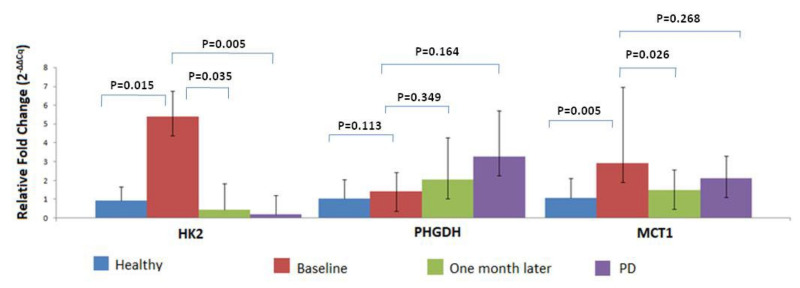
Relative fold change (2^−ΔΔCq^) of MRGs in CTCs from early stage NSCLC patient samples at different timepoints.

**Figure 4 cancers-14-03237-f004:**
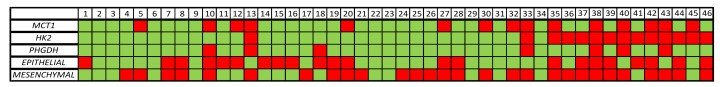
Heat map of epithelial markers (CK-8/CK-18/CK-19), EMT markers (VIM/TWIST-1) and MRGs (HK2/PHGDH/MCT1) as quantified by RT-qPCR. The red color represents overexpression, while the green color represents under expression or a lack of expression. The heatmap shows samples grouped by batch based on their positivity for MRGs.

**Table 1 cancers-14-03237-t001:** Clinicopathological characteristics of patients.

			*HK2 ^a^*	*p*-Value	*PHGDH ^a^*	*p*-Value	*MCT1 ^a^*	*p*-Value
Gender	Male	30	9	0.593	5	0.535	8	0.105
	Female	16	5		2		8	
Age	≤65	26	8	0.758	5		11	0.061
	>65	19	5		2		3	
	n.a.	1	1		1		1	
Stage	I	22	6	0.379	2	**0.040**	8	0.438
	II	10	4		3		3	
	III	13	3		1		3	
	n.a.	1	1		1		1	
Size	≤5	33	10	0.976	6	0.378	12	0.392
	>5	13	4		1		3	
Type	SCC	24	7	0.838	5	0.148	8	0.904
	ADENO	19	5		1		6	
	n.a.	3	1		2		1	
Smoking status	No	4	2	0.333	0	0.322	0	**0.017**
	Ex	17	3		1		2	
	Yes	20	7		4		10	
	n.a.	5	3		1		1	
Lymph Nodes	Yes	15	3	0.290	2	0.807	3	0.427
	No	31	11		5		11	

***^a^*** Columns depict the number of patients overexpressing the corresponding genes at baseline. The bold text highlights the significance of the test.

## Data Availability

All data generated or analyzed during this study are included in this published article (and its Appendix A).

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
