# Peer review of "Metabolism-Related Gene Expression in Circulating Tumor Cells from Patients with Early Stage Non-Small Cell Lung Cancer"

_cancers, 2022, doi:10.3390/cancers14133237_

Round 1

Reviewer 1 Report

Dear authors, 

I have read with interest you paper. I have a few observations:

  • In the introduction, the concept of metabolic investigations in CTC in NSCLC is defined. To my knowledge, the field has been actively investigated in last year. A rapid search in Pubmed revealed for example [Ziming Li, Nat Comm 2019; Turetta, Cancers 2018; Liu Yang, PNAS 2021], probably other studies will emerge from a thorough search. Please expand a little this part explaining how your study integrates into state of the art. 
  • Table 1: please add the description because data presentation is unclear (are numbers patients overexpressing genes in columns?). Also, the sum of patients does not result 46 in all rows (specifically, 45, 45, 46, 43, 41). Please review and if needed add the category n.a. for features described. 
  • 2.2. Even though I understand that this paper is not centered on the method of enrichment employed, the composition of the enriched sample is not clear. Is it checked/verified for having CTC? How? Does it have contaminants (non-CTC)? How many on average? Please add brief description of the method, with particular attention to the questions above.  
  • 3.1. Figure S1 is mentioned but it is not available to the reviewer
  • 3.3 line 201: how the associations are tested? Please define statistical test used.
  • Figure 4.
    • There is a pattern difference between patients 35-46, and patients 1-34. Are patients 35-46 the last one collected, in disease progression? This is interesting and if so should be highlighted.
    • If both epithelial and mesenchymal genes are not overexpressed, can we deduce that the sample is CTC-negative? This should be clarified. 
    • If only mesenchymal genes are overexpressed, how this ensures that CTC are contained in the sample? To my knowledge, detection of mesenchymal markers alone does not ensure that a malignant cells is detected in the sample. Other cells might be present, like connective tissue cells, either cancer-associated or not. This is the weakest point of the study and should be clarified, and highlighted in discussion. This is also related to the above comment on the purity of CTC/contaminants.
  • Discussion:
    • line 235-239: same observation of the introduction section: while mentioning pH regulation and glycolytic activity, it is worth mentioning other studies in literature which has investigated this in CTC. Please expand this section and describe how your study integrates and brings novelty to the field. Suggestions: [for pH: Zielke Anal Chem 2020, Brisotto Cancers 2020, Del Ben Ang Chem 2016, Rivello Sci Adv 2020; for glucose uptake Ziming Li, Nat Comm 2019; Turetta, Cancers 2018; Liu Yang, PNAS 2021, Venturelli Sci Rep 2016]
    • line 261: "all genes reduced one month after surgery", but in Figure 3 is shown that PHGDH does not. Please check.
    • Same paragraph: "while higher expression levels at the time of tumor relapse". I do not see it in Figure 3. Please check. 
    • A speculation is described about the association between glycolysis and mesenchymal phenotype. Besides clarifying purity of CTC in the mesenchymal subset, it might be interesting integrating into the discussion data from [Rivello Sci Adv 2020], in which a metabolic method is used to collect CTC and a majority of mesenchymal phenotype is detected. Authors conclude that majority of cells are not CTC but in my opinion data do not support this interpretation. Consider whether you want to integrate this study in the discussion. 

Minor corrections:

  • line 14: the acronym MRGs is used without prior definition
  • line 34: space is needed between . and Results:.
  • line 56: detect relapse, remove "of"
  • line 69: giveS, add S
  • line 127: perform, remove ED
  • line 129: cell line ) remove space between line and )
  • line 225: Alix-Panabieres is spelled incorrectly 

Author Response

In the introduction, the concept of metabolic investigations in CTC in NSCLC is defined. To my knowledge, the field has been actively investigated in last year. A rapid search in Pubmed revealed for example [Ziming Li, Nat Comm 2019; Turetta, Cancers 2018; Liu Yang, PNAS 2021], probably other studies will emerge from a thorough search. Please expand a little this part explaining how your study integrates into state of the art. 

We would like to thank the reviewer for this comment. As the metabolic phenotyping of CTCs have been evaluated by other studies, in the present study we focused on accessing gene expression (mRNA) of three crucial genes that associated with different metabolic pathways in the bulk of size-based CTC fraction. For the first time the change in expression levels of these genes was calculated at different stages of the disease in early stage NSCLC. The three genes of interest were carefully chosen in order to shed light for the first time in different, but yet associated metabolic pathways in the same samples. HK2 is indicative of the glycolytic phenotype of the cells, while MCT1 expression is evaluated in order to assess whether the isolated CTC utilize lactate under normoxic conditions, as it is known that this transporter is responsible for both the efflux and the uptake of lactate, and PHGDH is linked to Serine Synthesis Pathway (SSP), a branching pathway of glycolysis which might play a role in redox homeostasis and epigenetic modifications.”

In the revised manuscript we have added a paragraph which explains how our study integrates into state of the art.

Table 1: please add the description because data presentation is unclear (are numbers patients overexpressing genes in columns?). Also, the sum of patients does not result 46 in all rows (specifically, 45, 45, 46, 43, 41). Please review and if needed add the category n.a. for features described. 

In the revised manuscript on Table 1 we have added a description “Columns depict the number of patients overexpressing the corresponding genes. Bold: Highlights the significance of the test.” Moreover, based on the reviewer’s proposal we have added «n.a» category in the same Table.

2.2. Even though I understand that this paper is not centered on the method of enrichment employed, the composition of the enriched sample is not clear. Is it checked/verified for having CTC? How? Does it have contaminants (non-CTC)? How many on average? Please add brief description of the method, with particular attention to the questions above.  

Our results are based on bulk CTC analysis, so by using size-based microfluidic device we are actually not only enriching our samples with CTCs but we also co-isolate a low fraction of non-specifically PBMC. The presence of these non-specific cells is verified by expression of B2M in all our size-based CTC fractions. B2M expression is used as an internal control for sample quality to avoid false negative results, but also as a reference gene for relative quantification, since it is expressed in all cells, both CTCs and PBMC. Following this procedure and the identical analysis of Healthy donors peripheral blood samples we define a sample as CTC-positive based on the expression of these specific genes that differentiate CTCs from PBMC. In the revised manuscript we have added this information in section 2.2 and 3.3.

3.1. Figure S1 is mentioned but it is not available to the reviewer

We apologized for this omission. In the revised manuscript we have added the suppl.Figure 1.

  • line 201: how the associations are tested? Please define statistical test used.

We used Kruskal-Wallis (non-parametric) test to test whether the median expression levels are the same between these groups. In the revised manuscript we have added this information in the “2.6. Statistical Analysis” paragraph.

Figure 4: There is a pattern difference between patients 35-46, and patients 1-34. Are patients 35-46 the last one collected, in disease progression? This is interesting and if so should be highlighted.

This is an interesting observation but these patients are not the last collected. In this table we have group together the patients with most positive markers. This is why you noticed the difference pattern.

If both epithelial and mesenchymal genes are not overexpressed, can we deduce that the sample is CTC-negative? This should be clarified. 

We would like to thank the reviewer for this so important comment. If none of these genes are not expressed this is not mean that there is no CTCs in the tested fraction. High heterogeneity in gene expression in CTCs has been highlighted by many previous studies on bulk and single cell CTC analysis. Testing multiple RNA-markers on CTC we could increase the sensitivity of CTC detection. Based on these findings in the present study we have evaluated the expression of three epithelial markers and two mesenchymal markers but due to the heterogeneity of CTCs there are several other RNA markers that could be expressed in the size-based CTCs fraction. In the revised manuscript on Page 8 we have highlighted this observation.

If only mesenchymal genes are overexpressed, how this ensures that CTC are contained in the sample? To my knowledge, detection of mesenchymal markers alone does not ensure that a malignant cells is detected in the sample. Other cells might be present, like connective tissue cells, either cancer-associated or not. This is the weakest point of the study and should be clarified, and highlighted in discussion. This is also related to the above comment on the purity of CTC/contaminants.

As we described in above comment we evaluated the ‘background noise’ of leucocytes or other contaminants by analyzing peripheral blood samples from healthy individuals in exactly the same way as patients. We estimated a cut-off based on each gene normalized expression in respect to B2M expression in this control group. Using this approach we defined a sample as overexpressed (positive) based on the fold change of each gene expression in the CTC fraction in respect to the corresponding fraction in the group of healthy individuals.

Discussion:

line 235-239: same observation of the introduction section: while mentioning pH regulation and glycolytic activity, it is worth mentioning other studies in literature which has investigated this in CTC. Please expand this section and describe how your study integrates and brings novelty to the field. Suggestions: [for pH: Zielke Anal Chem 2020, Brisotto Cancers 2020, Del Ben Ang Chem 2016, Rivello Sci Adv 2020; for glucose uptake Ziming Li, Nat Comm 2019; Turetta, Cancers 2018; Liu Yang, PNAS 2021, Venturelli Sci Rep 2016]

We would like to thank the reviewer for this comment. In the revised manuscript we have added a paragraph in the discussion in order to summarize the findings of previous studies and to describe the novelty of our study.

“Recent studies exploited the metabolic reprogramming in an attempt to detect and isolate CTCs by label- free methods that are based on pH deregulation and acidification of the tumor microenvironment. Zielke et al presented an in-house droplet microfluidic technique which sorted cancer cells with different rates of glycolysis and enriched CTCs based on pH differences of single droplets [38]. Based on pH deregulations Brisotto et al detected metabolically-altered CTCs in metastatic breast cancer showing an association of elevated number of CTCs with a shorter overall and progression-free survival [39]. Recently, Yang et al demonstrated the possibility of characterizing a subpopulation of CTCs from NSCLC by HK2 as a metabolic function- associated marker, besides the epithelial marker- based detection approaches that are prone to miss a subpopulation of CTCs in the sample [40]. Our study exploits the metabolic reprogramming that happens in a tumor in an attempt to molecularly characterize size-based isolated CTCs according to their expression patterns, by studying at the same time three important genes that play a key role in the survival of cancer cells.”

line 261: "all genes reduced one month after surgery", but in Figure 3 is shown that PHGDH does not. Please check.

We apologized for this mistake in the revised manuscript we have corrected this sentence.

“Our results indicate that the expression levels of MCT1 and HK2 reduced one month after surgery compared to baseline, whereas expression levels started to rise again at the time of tumor relapse compared to one month after surgery in cases of MCT1 and HK2.”

A speculation is described about the association between glycolysis and mesenchymal phenotype. Besides clarifying purity of CTC in the mesenchymal subset, it might be interesting integrating into the discussion data from [Rivello Sci Adv 2020], in which a metabolic method is used to collect CTC and a majority of mesenchymal phenotype is detected. Authors conclude that majority of cells are not CTC but in my opinion data do not support this interpretation. Consider whether you want to integrate this study in the discussion. 

In the revised manuscript we have added a paragraph which further analysed the association between glycolysis and mesenchymal phenotype.

“Interestingly, we observed that the subpopulation of CTCs that exhibited both a glycolytic and a mesenchymal phenotype is significantly higher than that of glycolytic but not mesenchymal like CTCs. This can be explained by the fact that EMT is a trans differentiation process which requires a high amount of energy and a more rapid bio-synthetic rate to take place [46]. It is also shown that lactate as an end product of aerobic glycolysis, contributes to the alterations in the ECM (extracellular matrix) by acidifying it, that are favorable for invasion that leads to the formation of CTCs [47]. With that assumed, we can also consider that cancer cells that both express mesenchymal surface markers and exhibit a Warburgian metabolic phenotype are more likely to enter the circulation by intravasating, than those which have an epithelial phenotype and therefore are more tightly connected to each other. Rivello et. al, isolated highly metabolically active cells from the tumor microenvironment of prostate cancer patients, based on pH measurement. Interestingly they observed that the subpopulation of EpCAM- highly metabolically -cells was higher than that of EpCAM+ highly metabolically -cells. This is in line with our results, in which we observed a predominant glycolytic and mesenchymal-like subpopulation of the CTC fractions that we isolated. [48].”

Minor corrections:

  • line 14: the acronym MRGs is used without prior definition
  • line 34: space is needed between . and Results:.
  • line 56: detect relapse, remove "of"
  • line 69: giveS, add S
  • line 127: perform, remove ED
  • line 129: cell line ) remove space between line and )
  • line 225: Alix-Panabieres is spelled incorrectly 

In the revised manuscript we have corrected all these points.

Reviewer 2 Report

1. The authors provided table 1, which indicates the expression of HK2, PHGDH and MCT1, depending on the clinicopathological characteristics of patients. However, the gender of the patients is not specified, why is the threshold value for age chosen at 65 years? It is necessary to additionally indicate the status of the lesion of the lymph nodes. The data given in Table 1 are the results, but are given in the Materials and Methods section and are not discussed at all in the text. It is not clear whether the expression of HK2, PHGDH and MCT1 correlates with histological type, tumor size, lymph node involvement, etc. 2. There is no information about the group of healthy donors: gender, age, etc. 3. As for the tissues, it is also not clear what kind of tumors they are, no description is given.

Author Response

The authors provided table 1, which indicates the expression of HK2, PHGDH and MCT1, depending on the clinicopathological characteristics of patients.

However, the gender of the patients is not specified, why is the threshold value for age chosen at 65 years?

In the revised manuscript we have added a column in this table showing the gender of the patients. We have set the median age of our cohort as threshold.

It is necessary to additionally indicate the status of the lesion of the lymph nodes.

In the revised manuscript we have added in Table 1 the status of the lesion of lymph nodes.

The data given in Table 1 are the results, but are given in the Materials and Methods section and are not discussed at all in the text.

In table 1 we present the main patients’ characteristics and their correlation with the expression of MRGs in the baseline sample. Based on our results patients with positive MCT1 expression were found to be less frequently of non-smokers as compared to smokers (P=0.017) whereas no further significant associations were observed between MRGs expression and clinicopathological parameters. In the revised manuscript we have highlighted these findings in 3.3 section and in the discussion.

“Patients with positive MCT1 expression were found to be less frequently of non-smokers as compared to smokers (P=0.017). No further significant associations were observed between MRGs expression and clinicopathological parameters.”

“The correlation of MRGs overexpression in CTCs and patients’ clinicopathological parameters showed an association between the smoking status of patients and MCT1 overexpression, while there was no significant correlation between other patients’ clinicopathological parameters and any MRG. Domingo-Vidal et al observed in a previous study that cigarette smoke induces metabolic reprogramming of the tumor stroma in Head and Nech Squamous Cell Carcinoma (HNSCC), when cancer-associated fibroblasts cultured in a cigarette smoke extracts media were co-injected with HNSCC cancer cells [45]. Cigarette smoke induced oxidative stress, glycolytic flux and Monocarboxylate Transporter 4 (MCT4) upregulation contributing to tumor aggressiveness. We believe that further testing of MCT1 upregulation in NSCLC CTCs and smoking status in a larger cohort of patients would enlighten their possible correlation and the association with these patients’ clinical outcome[45]”

It is not clear whether the expression of HK2, PHGDH and MCT1 correlates with histological type, tumor size, lymph node involvement, etc.

As we mention above based on our results smoking is the only patient’s characteristic which significant associate with MCT1 overexpression, no further significant associations were observed between MRGs expression and clinicopathological parameters. In the revised manuscript we have highlighted these findings in 3.3 section and in the discussion.

There is no information about the group of healthy donors: gender, age, etc.

We would like to thank the reviewer for this notice. All HD had no known illness or fever at the time of blood draw, no history of malignant disease, were ≥35 years old, 52.6% were females and 47.4% were males. In the revised manuscript we have added these data in paragraph “2.1 Clinical Samples”

As for the tissues, it is also not clear what kind of tumors they are, no description is given.

We would like to thank for this comment. This training group of samples consisting of primary NSCLC (fresh-frozen) tissues and corresponding adjacent non-neoplastic tissues of 10 patients, all diagnosed with operable (stage I-III) NSCLC; tumor histology was squamous cell carcinoma (SCC; n = 3), adenocarcinoma (ADC; n = 7). In this group the majority of patients (60%) were smokers and their median age was 58 years. All patients were treatment naïve when the samples were collected. In the revised manuscript we have added this information in section “2.1 Clinical Samples”.

Round 2

Reviewer 1 Report

Comments are in the attached document. 

Author Response

In the reply to revision, authors said that “In the revised manuscript we have added a paragraph which explains how our study integrates into state of the art.”The added paragraph does not contain integration with state of the art nor any mention of previous studies, but only the explanation of the selected genes. Strangely though, in the reply to revision, Authors have written a sentence that address this comment, but they did not integrate it into the manuscript: “The metabolic phenotyping of CTCs have been evaluated by other studies [https://doi.org/10.1073/pnas.2012228118, ], Please check whether you missed to add this sentence to the manuscript, adding references for “other studies” or address the comment in another way.

We would like to thank the reviewer for this comment. In the revised manuscript in the last paragraph of introduction we have described more thoroughly why our study is a state of the art:

“Our lab was the first to demonstrate the overexpression of MCT4 in CTCs of early stage NSCLC patients. More specifically, we have shown the prognostic significance of MCT4 investigating its role as a potential non-invasive tumor biomarker [28]. As the metabolic phenotyping of CTCs has already been evaluated [29]–[31] and several groups have used metabolic features for CTCs isolation [32]–[34], in the present study we focused on accessing gene expression (mRNA) of three crucial genes that associated with different metabolic pathways in the bulk of size-based CTC fraction of early stage NSCLC patients. For this purpose, the metabolism related genes (MRGs) -HK2, PHGDH and MCT1- were carefully chosen and highly sensitive RT-qPCR assays were developed and validated for their relative quantification. The expression levels of MRGs were further investigated in size-based CTC fractions at different timepoints of the disease whereas their association with epithelial and EMT markers was also evaluated. Our findings indicate that metabolic function–associated markers are overexpressed in CTCs and should be prospectively evaluated as non-invasive circulating tumor biomarkers in a large and well-defined cohort of patients with NSCLC.”

 Supplementary Figure 1 is now both present as supplementary material and in the body of the manuscript. . Please provide a color legend (red – tumor; grey – normal). Y-axis is not labeled.

We apologize for this omission. In the revised manuscript we have corrected Y- axis label and added color descriptions and this figure presents only in Suppl.Files.

Figure 4. Ok, if so, this should be specified (samples are not chronologically ordered but grouped/ordered by some kind of clustering algorithm) to avoid this misinterpretation of data.

We would like to thank the reviewer for this comment. We have clarified this in legend.

Figure 3. Y-axis label lacks Delta Delta symbol.

In the revised manuscript we have corrected Y-axis label.

Figure 2 is not clear. What is Y-axis label? It cannot be Cq, tumor – Cq, normal, because data are also from adjacent non-tumor tissue. Is it Cq,target gene – Cq,reference gene? Also, numerical data reported in the text do not match Figure 2. E.g.: median PHGDH tumor “13.90” in the text while it looks like around “6” in the Figure. It would be nice to compare these values with values found in CTC in Figure 3.

In the revised manuscript we have corrected the Y-axis which is Cq, target metabolism gene – Cq, reference gene and the median values and we have corrected the median values.

Concerning the suggestion of the reviewer to compare CTC values with tissue values we would like to refer that these are not matched samples (CTC and tissues) of the same patients we believe that this point will confused the researchers. However based on the reviewer suggestion in the revised manuscript we have underlying the concordance of results between tissues and CTCs.

“Our analysis showed that both HK2 and MCT1 were also found to be overexpressed in the size-based CTC fractions of NSCLC patients whereas according to our findings, there was no concordance between the size-based enriched CTCs and paired primary tumors in respect PHGDH. A possible explanation for this finding could be based on tumor heterogeneity and the limited number of pair tissues.”

The question about quantification of contaminants remains without an answer, and this heavily affects interpretation of data. It is not possible to estimate real fold-expression without knowing composition of the sample. I will try to explain why with an example:

Let’s suppose that in our sample we have 995 contaminants and 5 CTC. Let’s suppose every cell has the same amount of B2M – 1 molecule; instead HK is 1 molecule for normal cells and 10-fold expression (10 molecules) for CTC. We will end up with:

  • 1000 molecules of B2M
  • HK 1 molecule * 995 normal cells + 10 molecules * 5 CTC = 995 + 50 = 1045, normalized to B2M à045-fold expression, which means that a real 10-fold difference would be undetectable.

Let’s consider instead 95 contaminants and 5 CTC, same parameters:

HK 1 molecule * 95 normal cells + 10 molecules * 5 CTC = 95 + 50 = 145 = 1.45-fold expression, which means that a real 10-fold difference would be heavily underestimated, if detectable.

Let’s consider instead 95 contaminants and 5 CTC with 100-fold difference:

HK 1 molecule * 95 normal cells + 100 molecules * 5 CTC = 95 + 500 = 595 = 5.95-fold expression, which means that a real 100-fold difference would be heavily underestimated, but most likely detectable.

For this reason, at least an average/range information on ratio between contaminants and CTC is needed for proper interpretation of results, and these calculations need to be showed somewhere, in results or discussion.

We would like to thank the reviewer for this so extensive and detailed description.

In the revised manuscript we have added this paragraph in order to clarify this comment.

“Our group have already shown that molecular assays based on real-time PCR car-ried out in nucleic acids material (RNA or genomic DNA) isolated from size-based CTC fraction [15], [28], [43] can give valuable information for the molecular characterization of CTC at the gene expression. It is clear that in this approach we are not verifying the presence of CTCs by imaging through immunofluorescence, but through the genetic material isolated size-based CTC fraction. This approach has been extensively and successfully used also for the molecular characterization of size-based CTC fraction by other research groups [44]–[46]. The microfluidics system that was used in this study assures the depletion of contaminating leucocytes for up to 106 fold providing a size-based CTC fraction with high purity and this better depletion of leukocytes reduce the RT-qPCR background and improve both specificity and sensitivity of the molecular approach [47]”

“To further increase the specificity of our developed assays for every marker showing a gene expression background in the healthy donor samples, a cut-off threshold value was calculated by adding the twofold standard deviation to the mean Ct (cycle threshold) value of these “false-positive” control samples.”

Finally, in our previous study we have performed experiments by spiking a known number of NCI-H1975 (10, 100, 1000 cells) in 10 mL peripheral blood (PB) of healthy donors, and we have showed that tumor cells were detected through an CK-19 mRNA expression in all cases.  It is known that CTC enrichment methods in peripheral blood samples always lead to a carried contaminant percentage of normal blood cells. To fully overcome this obstacle, the development of single-cell isolation and subsequent single-cell mRNA-seq assays is considered necessary and this is one of our future studies that we would like to do.  

Additionally – besides DeltaDelta-Cq, the absolute value of Cq must be showed in order to fully evaluate results.

In the revised manuscript we have added as Suppl.Figure 2 the absolute Cq values of each tested genes in the size-based CTC fraction of NSCLC patients and in healthy donors.

Additionally – considerations on how results compare with data shown on tumor tissue/adjacent tissue (Figure 2) must be made.

We have compared our results of tumor tissues with the results of TCGA. As we mentioned above these are not matched samples of peripheral blood and fresh tissue of the same patient and this is why we didn’t compare these results. However, based on the reviewer suggestion in the revised manuscript we have made a comment about this comparison.

“As our results clearly indicate that all tested MRGs are overexpressed in tumor tissues samples compared to adjacent tissues, we further proceeded in the evaluation of their expression levels in CTCs that shed from primary tumor. Our analysis showed that both HK2 and MCT1 were also found to be overexpressed in the size-based CTC fractions of NSCLC patients whereas according to our findings, there was no concordance between the size-based enriched CTCs and paired primary tumors in respect PHGDH. A possible explanation for this finding could be based on tumor heterogeneity and the limited number of pair tissues.”

Discussion: In the provided manuscript, part of the added references are in number format, and part in name format. The text in the manuscript does not match the text in the reply to revision, in which references up to number 48 are present, while only up to 44 in the manuscript. Please check this minor.

In the revised manuscript we have addressed this comment

But more importantly, the criteria employed in the selection of the references are not clear. Cited studies seems to be picked from the provided list without a rationale,and no additional studies are provided. The provided list was the result of a very rapid search in literature, which we consider a minimum requirement, to be expanded by proper literature search. At the same time, the integration of presented results with literature is scarcely described. Examples of lack of rationale: Authors cite studies assessing extracellular acidification, but not increased glycolysis – why? – results show increased HK2 which is directly connected to increased glycolysis. Authors cite studies from a different disease (Breast cancer, Brisotto) – which is good – but not studies from the same disease (NSCLC, Ziming Li and Turetta) – why?

Overall, please arrange discussion in a better and organic form.

Brief example:

CTC with abnormal metabolism have been investigated in NSCLC (refs), which confirm/integrate/do not agree with our results. Also, they have been detected in other diseases(refs), which strengthen/does not strengthen the fact that metabolism… . Several aspects of metabolism have been investigated, increased glycolysis (refs) and pH deregulation (refs), which are compatible/not compatible with our results (HK/glycolysis, MCT/extracellular acidification, …)

As an additional comment, the presented study is interesting and novel, and in particular two of the genes investigated are respectively involved in the two methods of investigation of abnormal CTC (HK – glycolysis, MCT – extracellular acidification) a little effort in depicting a more comprehensive vision of CTC biology emerging from collected evidence would increase the level of the study and is worth to be made.

The selection of references was based on the studies that their results were compatible with our study. In the revised manuscript we have added more clearly in the discussion the potential of each study.

“Our results indicate that the expression levels of MCT1 and HK2 reduced one month after surgery compared to baseline, whereas expression levels started to rise again at the time of tumor relapse compared to one month after surgery in cases of MCT1 and HK2.”

Look at Figure 3:

  • MCT1 reduced one month after surgery compared to baseline – ok
  • HK2 reduced one month after surgery compared to baseline – ok
  • MCT seems to rise again at tumor relapse – but no statistical test is shown between 1.46 and 2.1
  • HK2 does not rise again at tumor relapse – it goes down to 0.22 compared to 1.46

We apologize for the misinterpretation of data. In the revised manuscript (results) we have corrected the description of figure 3.

Reviewer 2 Report

I have no more comments/remarks on the article. I believe that in its present form the manuscript can be recommended for publication.

Author Response

We would like to thank the reviewer for positive comments

Round 3

Reviewer 1 Report

Introduction has been improved, but there are errors and omissions in citations.

In the introduction section, references are not corresponding:

-        As the metabolic phenotyping of CTCs has already been evaluated [29]–[31]”

Reference 29 actually does evaluate metabolic phenotyping (HK) describing a novel population of CTC,

Reference 30 does not evaluate any metabolic phenotyping of CTC. CTC are detected with immunophenotyping, and correlated to PET results from the primary tumor, but this is not a metabolic phenotyping of CTC.

Reference 31 does not evaluate any metabolic phenotyping of CTC. CTC are isolated by size and detected by morphological criteria. The study correlates counts with PET results from the primary tumor, but this is not a metabolic phenotyping of CTC.

At the same time, references [Ziming Li, Nat Comm 2019; Turetta, Cancers 2018], suggested in the previous review, that actually evaluate metabolic phenotype of CTCs (Ziming Li has “metabolic phenotyping” in the title!), are not cited in the whole document.

A thorough revision of references is warmly suggested.

-        Concerning the suggestion of the reviewer to compare CTC values with tissue values we would like to refer that these are not matched samples (CTC and tissues) of the same patients we believe that this point will confused the researchers.”

I understand that these samples are not matched, I suggested to make some considerations on the relative fold change found in tissue and CTC samples, in order to evaluate if values were in a comparable range, and if not authors should try to motivate it in the discussion. If you think that this would confuse researcher, please add a clarification, such as “samples are not matched, however, as a general consideration, we found a relative fold expression of x in tumor vs adjacent tissue and y in CTC vs normal tissue for MCT1….. etc…..”.

-        Values of HK2 in paragraph 3.2 are not shown, while PHGDH and MCT 1 are shown.

-        “Our group have already shown that molecular assays based on real-time PCR car-ried out in nucleic acids material (RNA or genomic DNA) isolated from size-based CTC fraction [15], [28], [43]……..

Maybe my comment has been misunderstood.

I am not questioning that PCR in enriched fraction can give valuable information.

I am not questioning that there are contaminants in the enriched fraction.

I am questioning that interpretation of results is not discussed properly. You need to inform the reader in a comprehensible way about what is the estimated real difference in gene expression between CTC and normal cells that you estimate, based on the measured difference in gene expression observed, taking into account the number of contaminants that you typically have in your enriched fraction. As I demonstrated in the comment, the measured difference does not correspond to the real difference by order of magnitudes, and this is not explained nor shown in the study, thus failing to describe the actual biological difference.

This comment from the following revision was not addressed: “For this reason, at least an average/range information on ratio between contaminants and CTC is needed for proper interpretation of results, and these calculations need to be showed somewhere, in results or discussion.

I am not asking for more experiments, but for elaboration of acquired data taking into account number of contaminants. This can largely improve your results, because a difference in gene expression measured in a contaminated sample is significantly underestimated with respect to the real difference.

-        To further increase the specificity of our developed assays for every marker showing a gene expression background in the healthy donor samples, a cut-off threshold value was calculated by adding the twofold standard deviation to the mean Ct (cycle threshold) value of these “false-positive” control samples.”

I do not understand where this point has been implemented, how does it change results or what does it mean. Do you mean that gene is considered/labeled as overexpressed/underexpressed only if Cq value is below/beyond mean+-2*SD with respect to healthy controls?

-        The discussion has been markedly improved, it is much more informative and comprehensive, congratulations.  

As a strengthening piece of evidence, I would suggest that at line 371, where you say that Yang et al. demonstrated that HK2 positive CTC were present in half of NSCLC patients, you mention that Turetta et al (Cancers 2018) showed that glucose-avid CTC are present in 85% of NSCLC patients, strengthening both your results and Yang study.

Overall I do not understand why authors in the current revision removed some references correctly added in the previous revision, anyway the discussion is in good shape now.

-        Supp. Figure 2 – it is not descripted whether displayed data are baseline, one month later, PD or all of these. Please specify. Also, there is no reason to display different symbols, scatterplot with one symbol only is sufficient. Boxplot with scattergram consistently with style of figure supp. 1 would be more appropriate, as it is difficult to understand distribution of values with superimposed datapoints.

-        Figure 3. Lacks error bars, please add. Also 3D-plot does not add information. A style consistent to Figure 2 would be much better.

-        Several language errors present throughout the document, especially in newly added sections, please double-check. E.g.:

317 have been showed à have showed

337 time or relapse à time of relapse

Round 4

Reviewer 1 Report

Figure 3 lacks error bars, please add them.

If you want to keep the Figure as it is in the body of text, please provide a supplementary figure with error bars. 
